# Optimal Selection of Thermal Energy Storage Technology for Fossil-Free Steam Production in the Processing Industry

**Anton Beck** [1], **Alexis Sevault** [2], **Gerwin Drexler-Schmid** [1], **Michael Schöny** [1] and **Hanne Kauko** [2,*]

1   Austrian Institute of Technology, Giefinggasse 4, 1210 Vienna, Austria; anton.beck@ait.ac.at (A.B.); gerwin.drexler-schmid@ait.ac.at (G.D.-S.); michael.schoeny@ait.ac.at (M.S.)
2   SINTEF Energy Research, Postboks 4761 Torgarden, 7465 Trondheim, Norway; alexis.sevault@sintef.no
*   Correspondence: hanne.kauko@sintef.no

**Abstract:** Due to increased share of fluctuating renewable energy sources in future decarbonized, electricity-driven energy systems, participating in the electricity markets yields the potential for industry to reduce its energy costs and emissions. A key enabling technology is thermal energy storage combined with power-to-heat technologies, allowing the industries to shift their energy demands to periods with low electricity prices. This paper presents an optimization-based method which helps to select and dimension the cost-optimal thermal energy storage technology for a given industrial steam process. The storage technologies considered in this work are latent heat thermal energy storage, Ruths steam storage, molten salt storage and sensible concrete storage. Due to their individual advantages and disadvantages, the applicability of these storage technologies strongly depends on the process requirements. The proposed method is based on mathematical programming and simplified transient simulations and is demonstrated using different scenarios for energy prices, i.e., various types of renewable energy generation, and varying heat demand, e.g., due to batch operation or non-continuous production.

**Keywords:** thermal energy storage; optimization; steam; power-to-heat; renewable energy

## 1. Introduction

Steam systems are a part of almost every major industrial process, in nearly all industrial sectors. Steam generation systems were estimated to account for 38% of global final manufacturing energy use or 44 EJ in 2005 [1], corresponding to 9% of the global final energy consumption. Steam production is still primarily based on the use of fossil fuels, and all the major industrial energy users devote significant proportions of their fossil fuel consumption to steam production [2].

There is thus an urgent demand to develop cost-efficient alternatives for fossil-based steam generation. Among these, thermal energy storage (TES) in combination with power-to-heat (P2H) conversion technologies such as electric boilers or high-temperature heat pumps (HTHPs) may enable a rapid transition towards renewables-based steam production with rather small changes in the infrastructure. Moreover, P2H combined with TES allows active participation of energy-intensive industries in the energy markets, which will be necessary for stable and flexible electricity supply in future decarbonized, renewables-based energy systems. At the same time, the industry can decrease its energy costs by shifting the electricity consumption to low-cost periods, and the security of supply can be increased.

Since short payback time and profitability are key criteria for investment decisions in the industry, it is necessary to identify cost-optimal integration scenarios for TES that also consider technical restrictions, such as available conversion technologies and thermodynamic constraints. Cost-optimal integration of TES has been studied in many different settings. Especially within the context of concentrating solar power plants, in combination with distributed energy systems, as well as in combined heat and power (CHP) and tri-generation

systems (combined cooling, heat and power—CCHP), cost optimal storage sizing and optimal operation are often addressed using mathematical programming techniques.

For example, for use in combination with a CHP unit, a sensible hot water storage model based on a network-flow model, which is a special case of linear programming model, was introduced [3]. The objective in this case was to optimize energy planning and trading within distributed energy systems, also targeting short-term trades at the spot market and participation at the reserve market providing balancing power. The DESOD (distributed energy system optimal design) tool is based on mixed-integer linear programming for optimal design and operation of distributed energy systems providing heating, cooling and electricity [4]. Within this tool, TES is considered using a capacity model (costs are driven by capacity, capacity is derived from the maximum energy content throughout the optimization period). Capacity models have also been used for the optimization of a tri-generation system including TES [5], within a simple storage model for optimization of a poly-generation district energy system [6], and for optimization including a simple ice storage with loss free heat transfer [7]. In the latter, the storage operates solely at phase change temperature and consists of a mixture of water and ice depending on the state of charge (SOC) of the storage.

Optimization performance and results for four different formulations for stratified TES using mixed integer linear programming (MILP) were investigated and compared to the widely used capacity models [8]. The authors showed that for their use-case, an energy system for building application, the capacity model overrates the system's efficiency and underestimates operating costs by 6–7%. Within a design methodology based on linear programming for designing and evaluating distributed energy systems, the authors use ideally mixed hot water tanks as thermal energy storage [9]. The storage thus shows a linear correlation between SOC and the storage temperature. Similarly, discrete temperature layers were introduced in a hot water storage tank model [10]. The model was used in a slave problem within an optimization strategy for district energy systems. A different approach was proposed for design optimization of a hybrid steam storage consisting of a Ruths steam storage combined with phase change materials (PCM) [11]. The problem was simplified by neglecting actual load requirements, but auxiliary parameters were introduced that account for different charging and discharging requirements.

Optimization models have also been used for operation optimization of TES. For the optimization of a CHP-based district heating system including TES with fixed size, upper and lower bounds for the SOC and also maximum charging/discharging rates were applied in order to maintain reliable operation [12]. The objective for this optimization model was to minimize energy acquisition costs. Dynamic programming was applied to find the optimal scheduling of power selling at the day-ahead market for solar thermal power plants with integrated TES [13].

In another work, the complex relations of design, operation and economics of solar thermal energy plants including the use of TES were studied [14]. In contrast to the works highlighted previously, dimensionless analysis was used in order to quantify TES efficiency.

Most of these approaches rely on predefined cost parameters, even though the actual TES requirements can have a significant impact on TES costs. Comparison of different TES technologies based on general KPIs is not possible, since performance of the individual storage depends significantly on various requirements (required temperature range, case specific restrictions, required heat loads, required capacities, etc.). For example, for Ruths steam storage, the applicable temperature range and especially the maximum allowable storage temperature and pressure both influence the volume and mass specific storage capacity in terms of energy content, but also the capacity-specific storage costs. The capacity-specific storage costs are the total storage costs per unit of energy content (e.g., €/kWh). Higher storage pressures not only result in thicker pressure vessels to contain increased internal pressures, but also reduce steel strength due to increased temperatures. Furthermore, load-dependent costs, which are especially important for TES systems that depend on heat transfer as a storage phenomenon, are often neglected. But it is obvious

that many storage technologies require components whose costs are driven by load, such as heat exchangers and pumps.

The present study proposes an optimization-based method for identifying the most cost-efficient TES system for load shifting and exploitation of fluctuating renewable energy sources in industrial steam production. The method considers case-specific TES requirements and accounts for heat load specific storage costs. P2H technologies and TES are combined to enable the interaction between thermal and electric energy systems, which allows the industry to actively participate in energy markets. The proposed methodology is demonstrated by different case studies representing different scenarios for electricity prices and process requirements such as temperature levels and dynamic heat demand.

## 2. Methodology

The goal of the proposed methodology is to obtain the optimal configuration of P2H systems for industrial steam supply which is selected from the superstructure shown in Figure 1. This not only includes the optimal storage capacity and the required heat loads but also optimal storage operation. The generalized methodology present in this work can be summed up as follows:

- Boundary conditions: heat demand, profiles for electricity costs, upper limit for steam supply temperature (steam generation) and lower limit for steam consumption (steam demand) temperature, maximum capacity and heat loads for cost functions generation (narrow limits increase accuracy of cost functions, but restrict solution space) are specified.
- Cost functions: for each TES technology, a cost function in terms of storage capacity and maximum heat load is obtained using cost data from a database of from the literature considering the most important cost drivers.
- Optimization model: the optimal combination of TES and steam-generation technologies, and their optimal operation is identified using a MILP/MIQP (mixed integer quadratic programming) model which is described in detail in Section 3.
- Recovery of storage details: after the optimal solution is calculated, TES specifications such as vessel size (volume, wall thickness), tube length, valves, etc. are recovered using technology-specific cost-function algorithms.

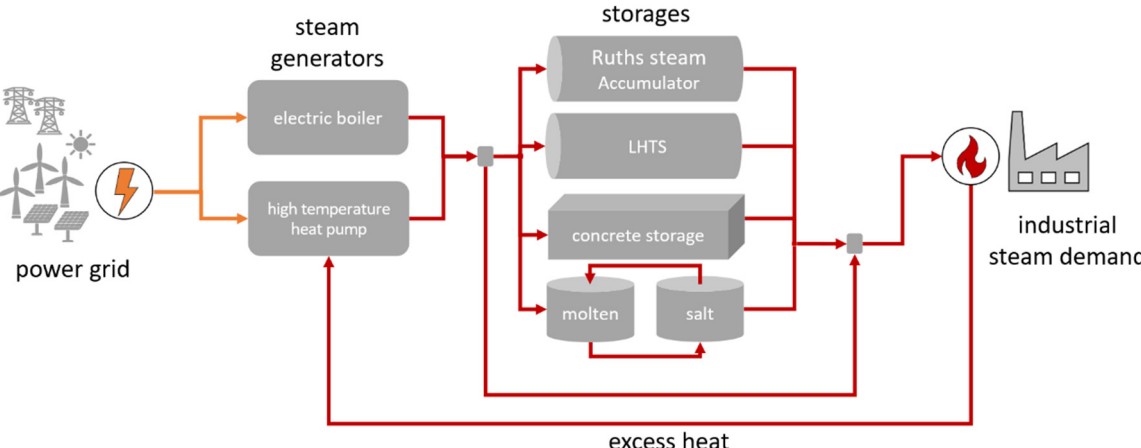

**Figure 1.** Schematic of the electricity-driven steam supply system considered within this work, showing the nodes and connectors considered in the model.

The storages in the optimization model are described with respect to capacity and heat load. From this, the detailed storage configuration is recovered with the algorithm used to obtain the storage cost-functions. The TES technologies considered in this work include:

- Ruths steam accumulators, which are the current state-of-the-art technology for steam storage [15]. Steam accumulators offer high charging/discharging rates, but the technology is limited by its relatively low energy density compared to e.g., PCM storage.
- Latent heat thermal energy storage (LHTS) using PCMs. LHTS offers high energy densities, and a temperature range that can be tailored to the application through optimal PCM selection [15]. However, the technology is still at a low TRL level and may suffer low heat-transfer rates.
- Sensible thermal energy storage in concrete, which offers a cost-efficient, safe and easy-to-use alternative for steam storage [16]. Limitations are low charging/discharging rates.
- Molten salt storages, which are widely applied in concentrated solar power [17]. Molten salts offer high thermal storage capacity and are also used as the heat transfer fluid (HTF). Limitations are corrosivity and high melting point temperature.

This selection of technologies covers a broad range of applications with regards to desired temperature level and charging/discharging rates and includes both state-of-the-art and emerging technologies. For steam generation, depending on the required steam quality, both electric boilers and HTHPs are considered.

## 3. Mixed Integer Linear Programming (MILP)/Mixed Integer Quadratic Programming (MIQP) Models

### 3.1. Electric Boilers

The optimization model for electric boilers considers the maximum heat load $\dot{Q}^{B,max}$ as the cost driver for investment costs and the required power $P_{el}^B$ as a driver for operating costs. The momentary heat load $\dot{Q}_t^B$ and the power consumption $P_{el,t}^B$ are linked through the boiler efficiency $\eta^B$. The index $t$ represents the operating periods and $NOP$ is the set of all these time periods.

$$\dot{Q}^{B,max} \geq \dot{Q}_t^B, \quad \forall\, t \in NOP \tag{1}$$

$$\dot{Q}_t^B = P_{el,t}^B\, \eta^B, \quad \forall\, t \in NOP \tag{2}$$

For simplicity, the investment costs for electric boilers $C_{invest}^B$ are considered to be a linear function of the maximum heat load $\dot{Q}^{B,max}$ with the cost coefficients $c_0^B$ and $c_1^B$.

$$C_{invest}^B = c_0^B + c_1^B\, \dot{Q}^{B,max} \tag{3}$$

Energy costs $C_{energy}^B$ are modelled as the sum of the momentary power consumption $P_{el,t}^B$ multiplied by the interval duration $\Delta t$ and the momentary electricity price $c_{el,t}$.

$$C_{energy}^B = \sum_{t \in NOP} \left( P_{el,t}^B\, \Delta t\, c_{el,t} \right) \tag{4}$$

### 3.2. High-Temperature Heat Pumps

Similarly, the heat pump model considers maximum heat load $\dot{Q}^{HP,max}$ as the cost driver for investment costs and the required power $P_{el,t}^{HP}$ as a driver for operating costs. The relation between the momentary HTHP heat loads $\dot{Q}_t^{HP}$ and its power demand is modelled using the Carnot equation and a heat pump efficiency $\eta^{HP}$:

$$\dot{Q}_t^{HP} = \frac{T_h}{T_h - T_c}\eta^{HP}P_{el,t}^{HP}, \; \forall\, t \in NOP. \tag{5}$$

The maximum heat load $\dot{Q}^{HP,max}$ is obtained using inequality constraints that force $\dot{Q}^{HP,max}$ to be greater than all momentary HTHP heat loads $\dot{Q}_t^{HP}$.

$$\dot{Q}^{HP,max} \geq \dot{Q}_t^{HP}, \quad \forall\, t \in NOP \tag{6}$$

The heat pump uses excess heat from the industrial process $\dot{Q}_{surplus,t}$ as a source. For simplicity reasons it is assumed that there is excess heat available only when there is a heat demand and that only a fraction of the process' heat demand is available as excess heat. It needs to be stated that this is generally not the case, especially for batch processing excess heat often occurs after heat is supplied to the batch. The proposed model can easily be modified if actual excess heat profiles are available to account for temporal differences between heat supply and excess heat availability. In addition, steam generation using HTHP is only feasibly if the required steam supply temperature $T_h$ is lower than the HTHP's maximum supply temperature $T_h^{max}$. Since HTHP do have limited sink temperatures, for this work, heat pumps are only considered up to a supply temperature $T_h^{max}$ of 160 °C.

$$\dot{Q}_t^{HP} - P_{el,t}^{HP} \leq \begin{cases} 0, & if\ T_h > T_h^{max} \\ \dot{Q}_{surplus,t}, & if\ T_h \leq T_h^{max} \end{cases}, \quad \forall\, t \in NOP \tag{7}$$

Just as in the case of electric boilers, the investment costs for the heat pump $C_{invest}^{HP}$ are considered to be linear and proportional to the maximum heat load $\dot{Q}^{HP,max}$.

$$C_{invest}^{HP} = c_0^{HP} + c_1^{HP}\, \dot{Q}^{HP,max} \tag{8}$$

Similarly, energy costs $C_{energy}^{HP}$ are calculated in the same way as for electric boilers (Equation (4)).

$$C_{energy}^{HP} = \sum_{t \in NOP} \left( P_{el,t}^{HP}\, \Delta t\, c_{el,t} \right) \tag{9}$$

### 3.3. Thermal Energy Storage

Even though different cost drivers need to be considered when it comes to the available TES technologies, in this work, the mathematical optimization models are based on the same constraints for each technology. The momentary energy content within the storage $Q_t^S$ is bounded by its upper and lower limits $Q^{S,max}$ and $Q^{S,min}$.

$$Q^{S,max} \geq Q_t^S \geq Q^{S,min}, \quad \forall\, t \in NOP \tag{10}$$

The usable storage capacity $\Delta Q^S$ is modelled as the difference between these upper and lower limits.

$$\Delta Q^S = Q^{S,max} - Q^{S,min} \tag{11}$$

The maximum charging $\dot{Q}^{S,max,c}$ and discharging heat loads $\dot{Q}^{S,max,d}$ are calculated by:

$$\dot{Q}^{S,max,c} \geq \dot{Q}_t^{S,in} - \dot{Q}_t^{S,out}, \quad \forall\, t \in NOP \tag{12}$$

$$\dot{Q}^{S,max,d} \geq \dot{Q}_t^{S,out} - \dot{Q}_t^{S,in}. \quad \forall\, t \in NOP \tag{13}$$

The current state of charge $Q_t^S$ is modelled recursively based on the previous time step and the incoming and outgoing heat loads. Cyclic operation is assumed and thus the SOC of the first and last timesteps are connected.

$$Q_{t=1}^S = Q_{t=NOP}^S + \left( \dot{Q}_{t=NOP}^{S,in} - \dot{Q}_{t=NOP}^{S,out} \right) \Delta t \tag{14}$$

$$Q_{t+1}^S = Q_t^S + \left( \dot{Q}_t^{S,in} - \dot{Q}_t^{S,out} \right) \Delta t, \quad \forall \, t \in \, NOP \tag{15}$$

Bounds for capacity $\Delta Q^S$ and heat loads $\dot{Q}^{S,max}$ are necessary to constrain the domain in the optimization problem to the same domain used for calculation of the cost functions.

$$\Delta Q^S \le \Delta Q^{S,max} \tag{16}$$

The heat load ratio $r$ is used to constrain the maximum heat load with respect to the actual storage capacity $\Delta Q^S$.

$$\Delta Q^S \, r \ge \dot{Q}^{S,max} \tag{17}$$

The binary variables $z^S$ are used to decide whether the storage is integrated.

$$\dot{Q}^{S,max} \le \Delta Q^{S,max} \, r^S \, z^S \tag{18}$$

For the LHTS, an appropriate PCM needs to be selected by the user. Since available PCMs have distinct melting temperatures, it might not be possible to use a PCM with equal temperature differences between the HTF and the melting temperature for charging and discharging. These potentially different charging and discharging behaviors are accounted for using charging and discharging efficiencies $\eta_c^S$ and $\eta_d^S$.

$$\dot{Q}^{S,max} \ge \dot{Q}^{S,max,c} \, \eta_c^S \tag{19}$$

$$\dot{Q}^{S,max} \ge \dot{Q}^{S,max,d} \, \eta_d^S \tag{20}$$

Depending on the selected accuracy of the approximate cost function, either a linear or a quadratic function is used to model the investment costs of the individual storage technologies $C_{invest}^S$ as a function of capacity and load. Usually, the cost functions somehow exhibit decreasing specific costs with the storage size and thus form non-convex functions.

$$C_{invest}^S = z^S * c_0^S + c_1^S \, \Delta Q^S + c_2^S \, \dot{Q}^{S,max} + c_3^S \, \Delta Q^S \dot{Q}^{S,max} + c_4^S \, \Delta Q^{S2} + c_5^S \, \dot{Q}^{S,max^2} \tag{21}$$

### 3.4. Excess Heat

As already mentioned in Section 3.2, the available surplus heat $\dot{Q}_{surplus,t}$ used as a source for HTHPs is limited and coexists with the processes' energy demand $\dot{Q}_{demand,t}$. The amount of surplus heat is modelled using a simple factor $f_{surplus}$ that describes which fraction of the heat demand is available as excess heat at a usable temperature level.

$$\dot{Q}_{surplus,t} = \dot{Q}_{demand,t} \, f_{surplus}, \quad \forall \, t \in \, NOP \tag{22}$$

### 3.5. Connectors and Nodes

To connect the selected TES and steam generators with the actual steam demand, two nodes are introduced to ensure the energy balance as shown in (23). Heat loads that by-pass the TES systems and are supplied directly to the process are accounted for as connector heat loads $\dot{Q}^C$.

$$\dot{Q}_t^{HP} + \dot{Q}_t^B = \dot{Q}_t^C + \sum_{i \, \in \, STO} \dot{Q}_{t,i}^{S,in}, \quad \forall \, t \in NOP \tag{23}$$

$$\dot{Q}_t^C + \sum_i \dot{Q}_{t,i}^{S,out} \ge \dot{Q}_{demand,t}, \quad \forall \, t \in NOP, \, i \in STO \tag{24}$$

*3.6. Objective*

The overall objective of the optimization model is to minimize the total annual costs $C_{total}$, which is a trade-off between investment costs for boilers, heat pumps and thermal storages on the one hand and energy costs on the other hand.

$$\min C_{total} = \underbrace{(C_{invest}^{HP} + C_{invest}^{B} + \sum_{i \in STO} C_{invest,i}^{S})}_{investment\ costs} f_a + \underbrace{C_{energy}^{HP} + C_{energy}^{B}}_{annual\ energy\ costs} \qquad (25)$$

To consider energy and investment costs on the same basis, the annualization factor $f_a$ is used to calculate annuities for the investments. In this case $f_a$ corresponds to the reciprocal of the equipment's life expectancy.

**4. Cost Functions**

The goal is to derive cost functions for the individual TES technologies that express total storage costs in terms of storage capacity and maximum heat load which can be used in the MILP/MIQP model presented in Section 3. For this reason, a predefined number of storage configurations in terms of geometries, thermal capacities and heat loads are calculated and evaluated. A detailed description for the technology-specific calculation of these configurations is presented in the following sections. Costs are calculated for every configuration using information from a cost database and from the literature. Suboptimal configurations in terms of total costs are eliminated. Suboptimal in this case means, that there are other storage configurations that have either at least the same maximum heat load at equal capacity but at lower total costs. A least squares fit is carried out for the remaining optimal configurations resulting in the desired cost function. In the case of a linear function the cost-function can be written as:

$$C_s = c_{s,0} + c_{s,1}C + c_{s,2}L, \qquad (26)$$

or in the case of a quadratic function

$$C_s = c_{s,0} + c_{s,1}C + c_{s,2}L + c_{s,3}CL + c_{s,4}C^2 + c_{s,5}L^2, \qquad (27)$$

where $C_s$ is the storage costs, $C$ is the storage capacity, $L$ is the maximum storage heat load and $c_{s,1...5}$ are the cost coefficients.

The equipment considered within the individual cost functions and the parameters that impact the specific cost drivers is listed in Table 1.

**Table 1.** Components and key variables that impact the respective component costs for the selected thermal energy storage (TES) technologies.

| | | Ruths Steam Storage | Latent Heat Thermal Energy Storage | Molten Salt Storage | Concrete Storage |
|---|---|---|---|---|---|
| Heat storage material | PCM, salt, concrete | - | max./min. temperature, volume | volume | volume |
| Steel tubes [18] | Seamless, stainless steel | - | tube diameter, tube length | - | tube diameter, tube length |
| Steel plates [18] | S234JR | - | surface area | - | - |
| E-motors [19] | - | - | - | heat load | - |
| Pumps [18] | Single stage, cast iron | - | - | heat load | - |
| Vertical storage tanks [18] | Cone roof, carbon steel | - | - | volume | - |
| Cylindrical storage vessels [18] | Carbon steel | volume, required wall thickness | - | - | - |
| Heat exchangers [18] | U-Type, Stainless steel | - | - | heat load | - |

**Table 1.** *Cont.*

|  |  | Ruths Steam Storage | Latent Heat Thermal Energy Storage | Molten Salt Storage | Concrete Storage |
|---|---|---|---|---|---|
| Thermal insulation [18] | Glass wool with aluminum sheeting | max. temperature, surface area | max. temperature, surface area | max. temperature, surface area | max. temperature, surface area |
| Valves [a] | depending on TES type | max. temperature, heat load | Fixed value per container unit | Fixed value per storage unit | Fixed value per container unit |

[a] Spirax Sarco SV 60.

### 4.1. Ruths Steam Accumulators

The main cost driver for Ruths steam storages is the pressure vessel. The maximum temperature range from $T_{min}$ to $T_{max}$ is discretized in $n$ equidistant steps. Volume specific thermal storage capacities are calculated for given operating temperature ranges from $T_{min}$ to $T_{max,n}$ for a given maximum filling level of the pressure vessel $f_0$. The calculations are performed using the Coolprop Wrapper [20] for fluid properties in Python. The vessel is initialized at $T_{max,n}$ with $f_0 = f_{max}$. All steam inside the pressure vessel is extracted and the new equilibrium is calculated. This step is repeated until the storage temperature drops below $T_{min}$ which terminates the simulation. The total extracted energy yields the volume-specific storage capacity for a given operating temperature range and the maximum filling level $f_0$. The procedure to calculate the storage capacity for given minimum and maximum temperatures is presented in Figure 2 (left).

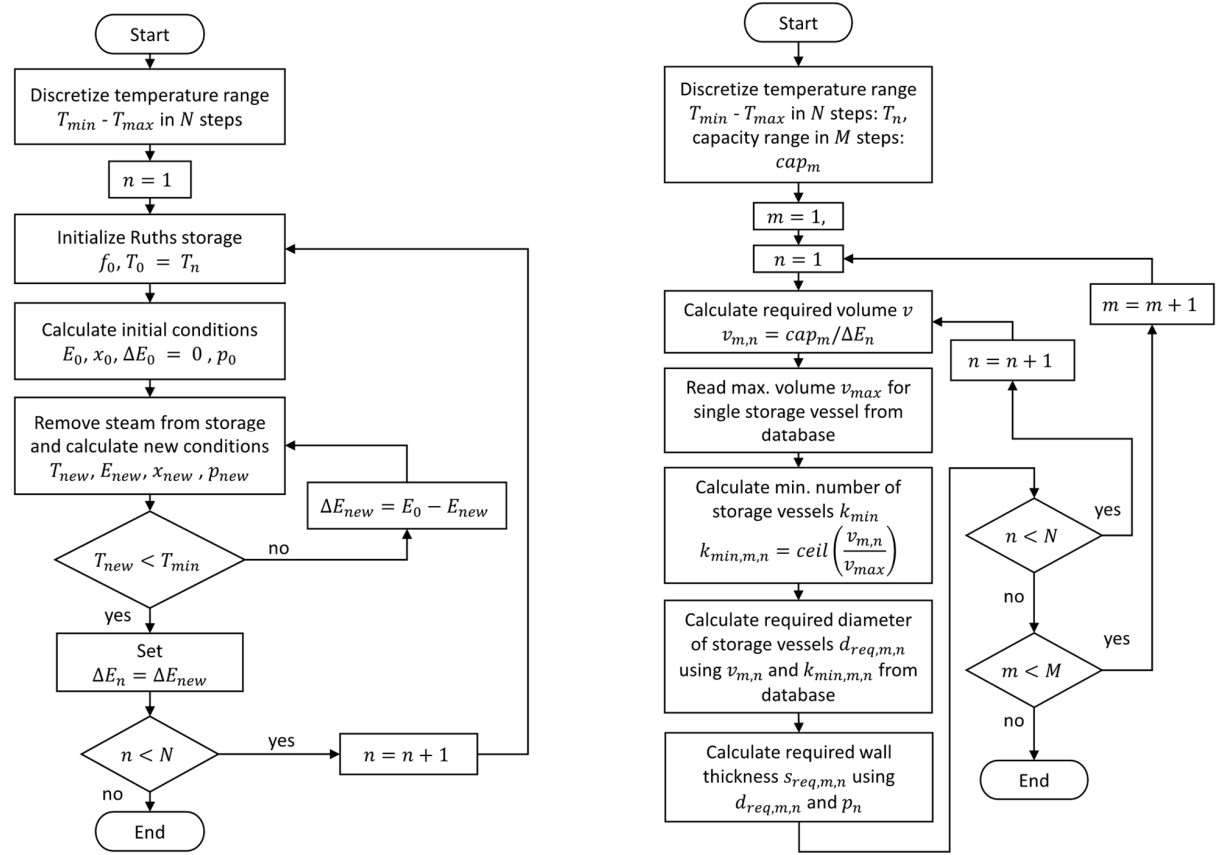

**Figure 2.** Ruths steam accumulator: calculation of vessel capacities (**left**) and calculation of storage parameters (**right**).

Now, for each $T_{max,n}$, the required vessel volume, the number of storage vessels and the required wall thickness is evaluated for user-defined discrete values of thermal storage

capacity (Figure 2 (right)). The required wall thickness is calculated according to any pressure vessel norm such as DIN EN 13,445 or the ASME (American Society of Mechanical Engineers) code. For this work, the AD 2000 norm [21] was used to calculate the necessary wall thickness.

The total vessel costs are then calculated using costs from a cost database for cylindrical pressure vessels [18]. Since only discrete volumes and wall thicknesses are available on the market, costs for the required storage parameters are either interpolated or the next larger vessel with suitable properties is selected. If the available storage volumes are not sufficient, multiple storage vessels are selected. Insulation costs for the pressure vessels are calculated using a correlation based on equipment temperature and equipment factors accounting for special insulation requirements.

Piping needs to be selected according to required flow rates. In this work, the maximum flow rate within the inlet and outlet of the vessel is set to 25 and 20 m/s, respectively. This is slightly lower than the limits of 25 m/s for saturated steam (outlet) and 40–60 m/s for dry steam (inlet) as suggested in literature [22]. Several valves are needed in a steam accumulator (see Table 2), and the valves are selected according to the required piping diameters to satisfy the velocity limits. Maximum flow rates are discretized from 0 to $\dot{Q}_{max}$ and, depending on the maximum temperature, are converted to mass flows. These mass flows are then used to identify required pipe diameters for the outlet and inlet of the storage.

**Table 2.** Valves and instrumentation considered for Ruths steam storage. Prices are according to [18,23,24].

| Type | Quantity Per Storage (pcs.) | Total Costs (€) |
|---|---|---|
| bourdon pressure gauge incl. ring type syphon tube, liquid damping | 3 | 1260.- |
| bimetallic temperature gauge incl. thermo wells | 3 | 1455.- |
| Drain valve DN50 PN40 | 1 | 830.- |
| Vacuum breaker DN15 PN40 | 1 | 340.- |
| Relief valve | 1 | * |
| Pressure reducing valve | 1 | * |
| Safety valve | 1 | * |
| Float ball valve | 1 | * |

* calculated for each storage configuration, depends on storage requirements.

### 4.2. Latent Heat Thermal Energy Storage (LHTS) and Concrete Storage

Both the LHTS system and the concrete storage considered in this work consist of a tube bundle surrounded with thermal storage material, as shown in Figure 3. For both charging and discharging, the heat transfer fluid flows through the same tubes. It is assumed that the heat transfer fluid is liquid water or steam, respectively. When the thermal storage is charged, steam flows through the pipes and condenses, whereas in the case of discharging, liquid water evaporates within the tubes. It is assumed that the mass flow of the heat transfer fluid is controlled to ensure full evaporation or condensation within the storage tubes.

Figure 4 (left) shows the flow-chart for the calculation of the different storage configurations for LHTS and concrete storages. The tube diameter $d_{tube}$ and the heat storage material layer $s_{mat}$ are varied within user-defined ranges. For each combination of tube diameter and storage material layer a charging cycle is simulated. Since the dynamic behavior of the concrete storage and even more so of the LHTS is highly complex and a rigorous transient simulation model would result in excessively long computation time, a simple quasi-stationary node model illustrated in Figure 5 using the so-called enthalpy approach is used for simulation.

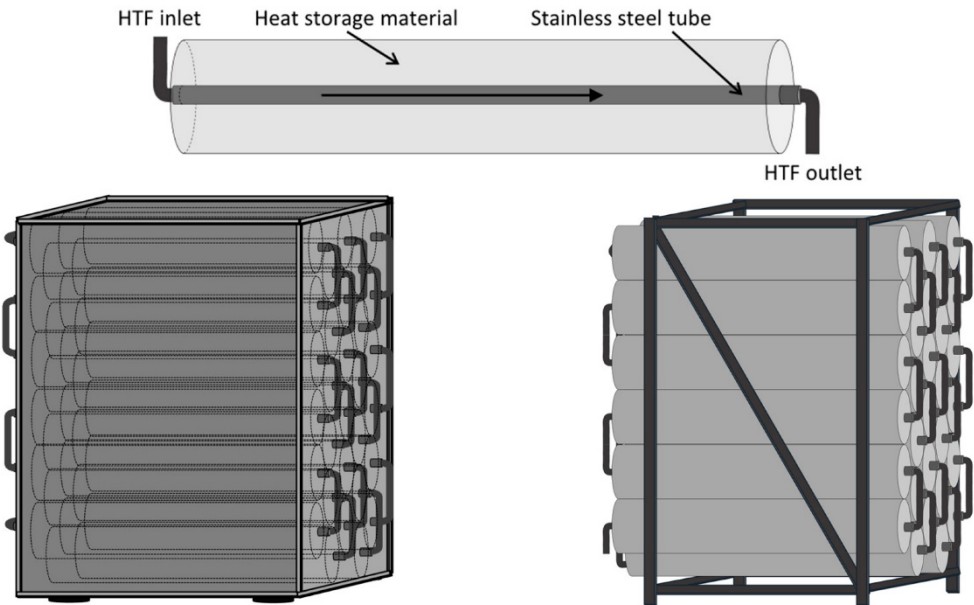

**Figure 3.** Schematic drawings of the tube surrounded by heat storage material for both LHTS and concrete storage (**top**), the LHTS system (**left**) and the concrete storage system (**right**) considered in this work. Both TES systems are represented without thermal insulation material.

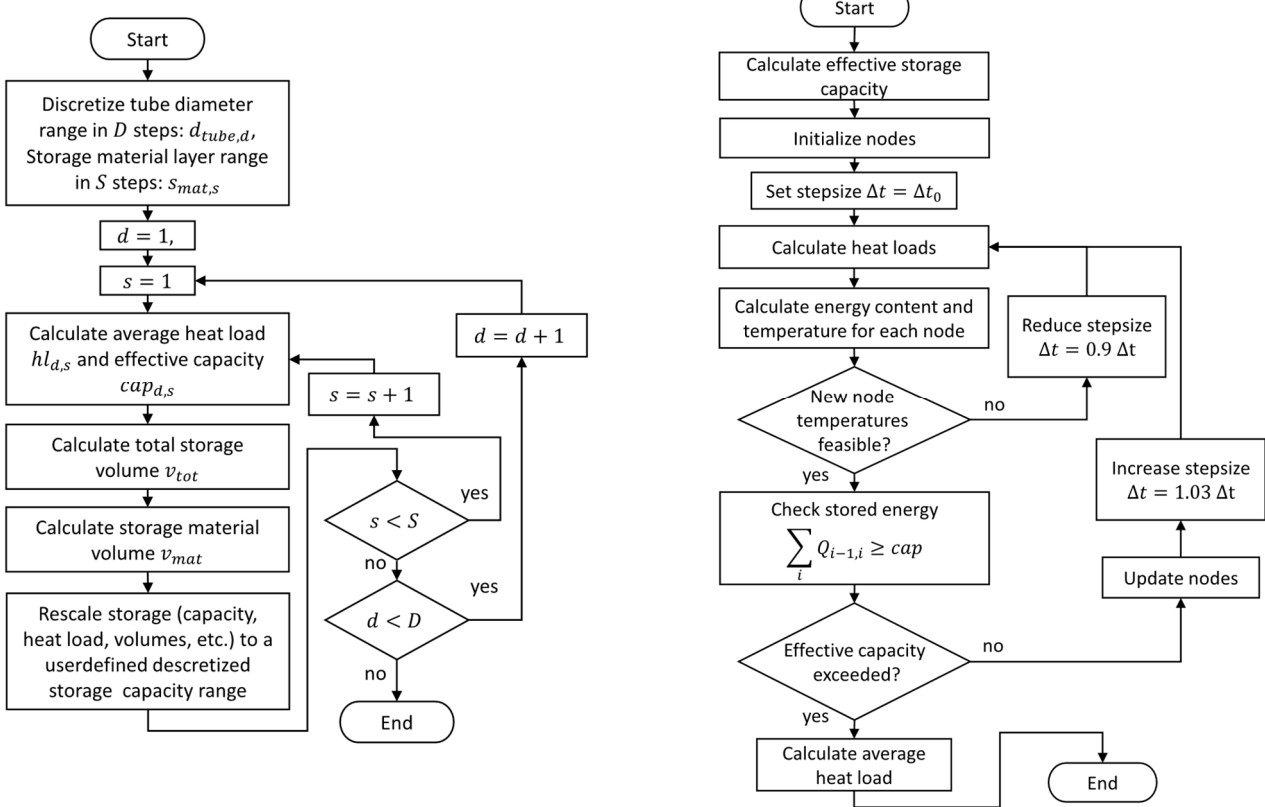

**Figure 4.** Flow-charts for the calculation of storage parameters for LHTS and concrete storages (**left**) and for the calculation of average heat loads (**right**).

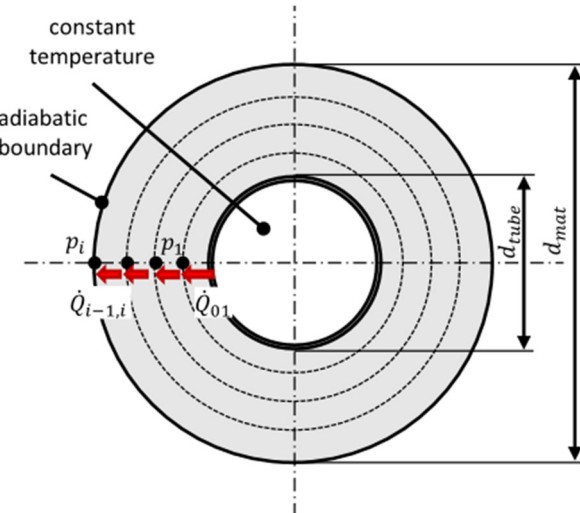

**Figure 5.** Schematic of the node model for LHTS and concrete storage.

In this model, the storage material layer is divided to discrete volumes with index $i$. These volumes are defined by:

$$v_i = \left( \left( \frac{d_i}{2} \right)^2 - \left( \frac{d_{i-1}}{2} \right)^2 \right) \pi \, l, \quad d_{i-1=0} = d_{tube}. \tag{28}$$

To account for the fact that a sufficient temperature difference between storage material and HTF is necessary to obtain sufficient heat loads, an effective temperature range is specified that depicts the useful temperature range for storage of sensible heat. For LHTS, the total storage capacity $cap_{total}$ considering the effective temperature range $\Delta T^{eff}$ is calculated: by

$$cap_{total} = v_{mat} \left( h_{lat} + c_p \, \Delta T^{eff} \right). \tag{29}$$

whereas for concrete, the storage capacity calculation simplifies to:

$$cap_{total} = v_{mat} \, c_p \, \Delta T^{eff} \tag{30}$$

with

$$\Delta T^{eff} = (T_{max} - T_{min}) \eta_T \tag{31}$$

where $\eta_T$ is the temperature efficiency factor, which was set to 0.8 in this work. This factor reduces the theoretically available temperature range to a more realistic range where reasonable driving temperature differences are ensured. The heat transfer between HTF and the heat storage material is governed by:

$$kA_0 = \alpha \, d_{tube} \pi \tag{32}$$

and the $kA$-value for heat conduction between the nodes is:

$$kA_i = 2 \frac{\lambda \pi}{\log \left( \frac{d_i}{d_{i-1}} \right)}. \tag{33}$$

The HTF remains at constant temperature $T_0 = T_{max}$ since a phase change between liquid water and steam takes place. The simulation is initialized with homogenous temperatures throughout all nodes and stored energy is set to zero.

$$T_{i,t=0} = T_{min} + \frac{(T_{max} - T_{min})(1 - \eta_T)}{2}, \quad \forall i \in I. \tag{34}$$

$$Q_{i,t=0} = 0, \quad \forall i \in I. \tag{35}$$

The simulation is then carried out using an initial step size $\Delta t$ which is adjusted if the current step results in an infeasible solution for the node temperatures. First heat loads $\dot{Q}_{i-1,i,t}$ are calculated,

$$\dot{Q}_{i-1,i,t} = kA_i\,(T_{i,t} - T_{i-1,t}), \qquad \dot{Q}_{0,1,t} = \frac{1}{\frac{1}{kA0} + \frac{1}{kA_1}}(T_{1,t} - T_0) \tag{36}$$

then the stored energy $Q_{i,t}$ is obtained by:

$$Q_{i,t} = Q_{i,t-1} + \left(\dot{Q}_{i-1,i,t} - \dot{Q}_{i,i+1,t}\right)\Delta t. \tag{37}$$

In the concrete storage case, the new node temperature is obtained through

$$T_{i,t} = \frac{Q_{i,t}}{v_i c_p} + T_{i,t=0},. \tag{38}$$

whereas for the LHTS also the current state of the PCM needs to be identified in order to determine the node temperatures.

$$T_{i,t} = \begin{cases} \frac{Q_{i,t}}{v_i c_p} + T_{i,t=0}, & if\ Q_{i,t} < Q_{sl} \\ T_{melt}, & if\ Q_{sl} \le Q_{i,t} < Q_{ll} \\ \frac{Q_{i,t} - v_i h_{lat}}{v_i c_p} + T_{i,t=0}, & if\ Q_{i,t} \ge Q_{ll} \end{cases} \tag{39}$$

$$Q_{sl} = (T_{melt} - T_{i,t=0})v_i c_p, \text{ and } Q_{ll} = (T_{melt} - T_{i,t=0})v_i c_p + v_i h_{lat}. \tag{40}$$

From these results, the average storage heat loads are derived. Since at the beginning of each charging and discharging cycle, heat loads are very high but only for a short period of time, these high charging rates are not considered for the calculation of average heat loads. Since for this simple model heat loads scale linearly with capacity (tube length), all solutions can be upscaled to discrete capacities ranging from 0 to the user specified maximum capacity.

For the LHTS, an appropriate PCM needs to be selected by the user. The most important property is the phase change temperature, which needs to be between the charging and discharging temperature of the HTF. Besides costs for the PCM itself, which strongly depend on the selected PCM as shown in Figure 6, PCM selection has various implications on storage costs. PCMs with low densities result in larger overall storage volumes and, depending on phase change enthalpy, lower volumetric energy densities, which in turn also requires larger surface areas between tubes and PCM to reach certain heat loads. For this reason, LHTS costs can vary significantly depending on its application in terms temperature range of operation.

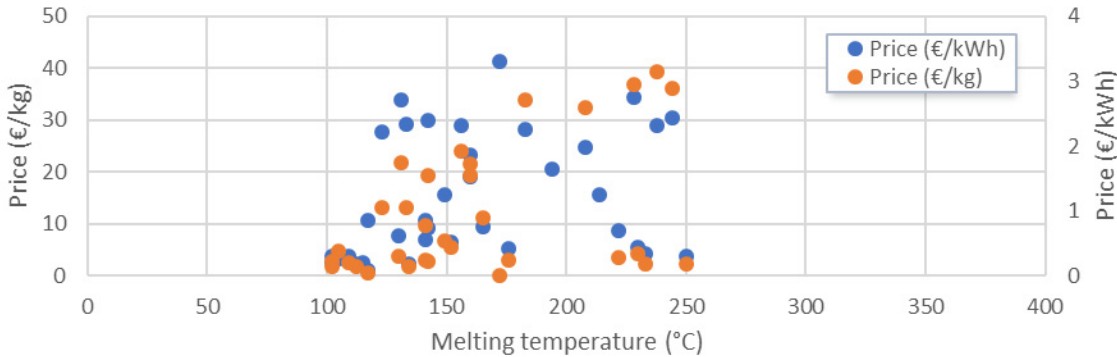

**Figure 6.** Price ranges for PCM in terms of €/kg and €/kWh (based on [25]).

The price for thermal concrete is not available in the literature. However, it is within the highest range of concrete available on the international market, since concrete used for concrete-based TES shall have specific thermodynamic and mechanical properties to perform durably and effectively. Considering an average price of 124 EUR/m$^3$ in 2018 for dry concrete (National Ready Mixed Concrete Association—NRMCA—Industry Data Survey 2018), a rounded price of 200 EUR/m$^3$ dry concrete (ca. 60% above the mentioned average) was assumed in this work to account for the specificities of the thermal concrete.

For each storage configuration, an appropriate storage container is selected. For the LHTS system steel plates are considered to encapsulate the PCM, whereas for the concrete storage system, the tube bundle arrangement does not require any containing vessel since the concrete surrounding the tubes will remain solid and contain itself. A simple metallic structure can hold the tube bundle together. The proposed structure is similar to the configuration proposed by EnergyNest for their pre-commercial concrete TES system [16].

For both LHTS and the concrete storage, thermal insulation is used around the container and the metal structure, respectively. Insulation costs are calculated using a correlation based on equipment temperature and equipment factors accounting for special insulation requirements. Costs for valves and sensors are based on equipment purchases from previous projects and are presented in Table 3.

**Table 3.** Costs for valves and sensors for LHTS and concrete storage based on previous projects.

| Type | Vendor | Quantity (pcs.) | Costs Per Storage Unit (€) |
|---|---|---|---|
| Temperatures sensors PT-100 | www.jumo.com | 2 | 800 |
| Ultrasonic flow meter | www.flexim.com | 1 | 500 |
| Thermocouples | www.tcdirect.de/ | 20 | 1000 |
| Valves | www.ari-armaturen.com/ | 2 | 1000 |

### 4.3. Molten Salt Storage

The molten salt storage was modeled as a conventional two-tank solution with one hot tank and one cold tank, as illustrated in Figure 7 (left). The hot tank and cold tank temperatures were set equal to $T_{max}$ and $T_{min}$, respectively. The thermal storage is charged with steam via a heat exchanger and discharged similarly by reversing the flow. The cost function for molten salt storage thus includes the costs for heat storage material, storage tanks and insulation, heat exchangers, pumps and electric motors. Of these, the costs for pumps, electric motors and the heat exchanger depend only on heat load, whereas the costs for the remaining components depend only on thermal storage capacity. Figure 7 (right) illustrates the approach for calculating the required salt volume and flow rate, and consequently the required sizes for heat exchangers, pumps and electric motors are calculated for each capacity and load in the specified range.

As the heat storage material, a novel ternary salt mixture called Yara MOST, which is a blend of $Ca(NO_3)_2$, $KNO_3$ and $NaNO_3$, was considered [26]. The benefits of Yara MOST as opposed to other salts applied in concentrated solar plant (CSP) applications are among others its low melting point (131 °C) reducing the risk of freezing, wider operational temperature range, almost no corrosion and lower cost. The use of Yara MOST as a heat transfer fluid and TES medium has been tested at industrial scale at a parabolic trough CSP plant in Portugal [27]. A constant price at the lower limit obtained from the supplier, equal to 0.7 €/kg, was applied for the salt. Reduction in price due to increased quantity was not considered due to lack of data.

Due to the low corrosivity of the salt, and generally low temperatures employed in industrial applications, carbon steel was considered as the tank material. Since the storage tanks are under atmospheric pressure, the tank thickness was set to a constant value of 10 mm, even though in certain cases thicker walls might be necessary. The costs and required number of tanks were subsequently obtained from a cost database for vertical storage tanks [18], with the required salt volume as the input parameter. Similarly, the tank

insulation costs were obtained from the cost database, with maximum tank temperature and surface area for each tank as input.

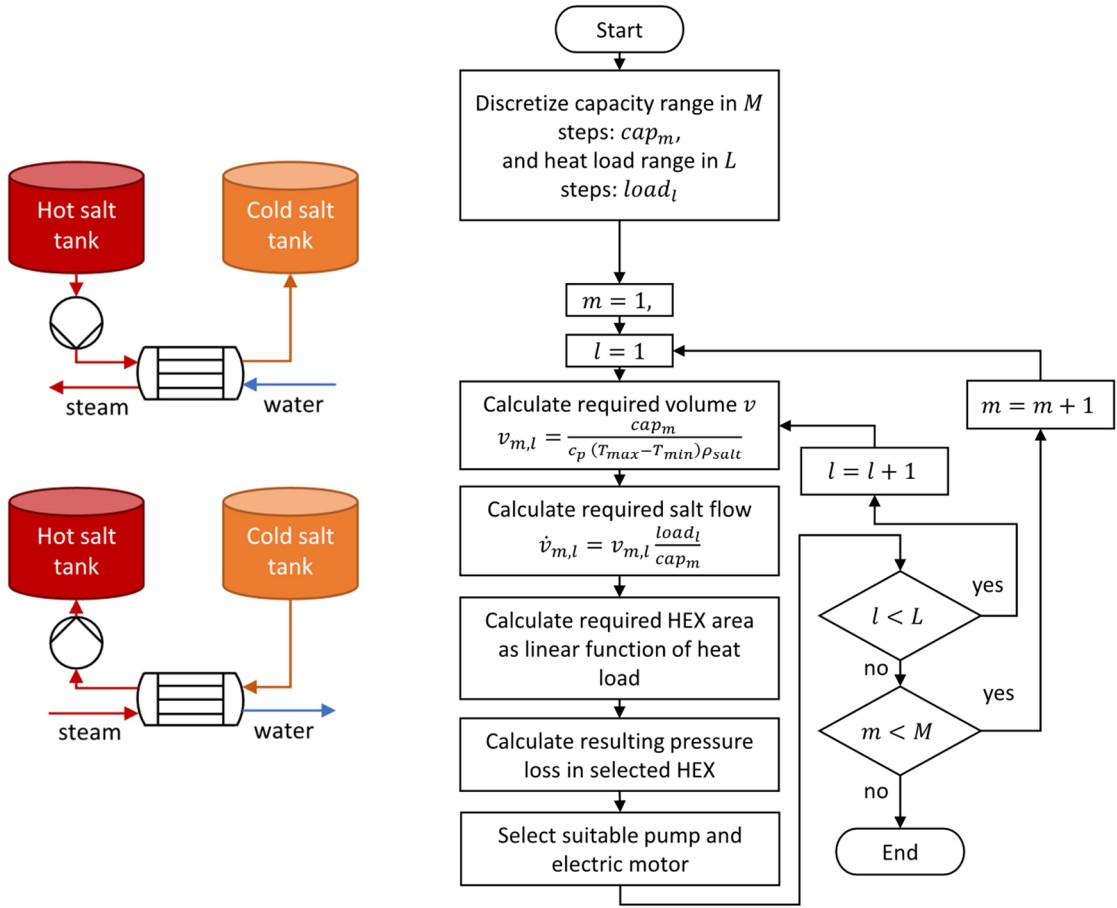

**Figure 7.** Sketch of the molten salt TES system in charging and discharging modes (**left**) and calculation of storage parameters and selection of pumps and motors for molten salt storage (**right**).

Molten salt steam generators generally consist of several heat exchanger steps [28,29]. For the present study, only the evaporation stage was considered in order to be consistent with the other storage technologies. The evaporator was assumed to be a U-type stainless steel heat exchanger with water flowing in the tubes and salt in the shell side. For calculating the heat transfer coefficient for water in the evaporator, the Gungor and Winterton correlation was applied [30]. For the heat transfer coefficient for the salt flowing across the tube bundle, the approach given by Gnielinski [31] was followed, assuming a staggered tube arrangement and a triangular pitch with $P_t = 1.25d_o$, with an outer tube diameter $d_o$ of 0.023 mm.

The overall heat transfer coefficient and thus the required heat transfer area was calculated for a range of loads and numbers of tubes, $N_{tubes}$. The tube bundle diameter was calculated from basis of the number of tubes using correlations given in [32], and the shell diameter was estimated to be 1.1 times the bundle diameter. From the range of obtained heat transfer areas, only those that satisfied the following condition were considered [32]:

$$D_{shell} < L_{tube} < 10D_{shell} \tag{41}$$

where $D_{shell}$ is the shell diameter and $L_{tube}$ is the length of a tube. For each load, the minimum heat transfer area satisfying this condition was selected. Finally, using the selected heat transfer areas, a linear function for the area as a function of load was obtained to be applied in the optimization model in order to minimize the computation time. The

same procedure was applied for obtaining the required number of tubes for each load, which was needed in calculating the pressure drop as explained in the following section.

The cost function for the salt pump was obtained using the cost database with salt flow rate and pressure drop as the input parameters. The largest pressure drop will take place in the heat exchangers, and the required pump size was thus estimated based on this pressure drop, calculated from [33]

$$\Delta p = N_L \chi \, f \, \frac{\rho v^2}{2} \tag{42}$$

where $N_L$ is the number of tube rows, estimated as $\sqrt{N_{tubes}}$, $\chi$ is a correction factor set to 1, $f$ is the friction factor, $\rho$ is the average salt density, and $v$ the flow velocity. The friction factor was set equal to the Euler's number, calculated from the Reynolds number of the flow using correlations given in [34].

An electric motor is needed for running the pump, with size and efficiency depending on the salt volume flow, i.e., the load. The electric motor efficiency and the costs were calculated using correlations found in [19].

### 4.4. Steam Generator Units

Since the focus of this work is on the development of reliable cost estimates for thermal energy storage, costs for steam generator units are modelled using linear correlations with respect to the components' nominal heat loads. The cost coefficients for these linear correlations are based on experience and are to be considered as rough estimates.

## 5. Example Cases

Two cases with very different characteristics were selected to demonstrate the presented approach for cost optimal integration of thermal energy storage and to highlight its capabilities.

### 5.1. Example Case 1—Large-Scale Plant with Constant Steam Demand and High Temperature

Case 1 represents a very large industrial facility with a constant steam demand of 1200 t/h which corresponds to about 900 MW. Steam needs to be supplied at 200 °C and can be produced at 300 °C saturated steam. The facility is located near the Equator and thus the year is split into dry season and wet season, which is reflected in the electricity prices as a large share of the power production is based on hydropower. For each season, one representative week was selected and was repeated for half-a-year. Energy prices for the two representative weeks are presented in Figure 8.

The cost structure for all considered storage types is presented in Figure 9 considering the thermal requirements of Case 1. For the LHTS with $KNO_3$-$NaNO_3$ as a PCM at 1000 €/m$^3$, the storage material costs dominate the overall costs for each application area. Concrete storage shows a similar cost structure however, storage material costs make up for a lower share of total costs. For both LHTS and concrete storages the share of tube costs increases with heat loads for both storage types since larger heat transfer areas are required. Costs for Ruths storages are dominated by vessel costs which make up for more than 85% of the overall costs for each dimensioning range. In contrast to the other storage types where valve costs are negligible, valve costs for Ruths add up to about 10%. Similar to LHTS and concrete storage the storage material costs dominate the overall costs for molten salt storage with a share of over 85%, followed by vessel costs in all dimensioning ranges. All other cost drivers combined are in the range of <5%.

The optimal system for Case 1 is shown in Figure 10 and is summarized in Table 4. It consists of an electric boiler with a maximum load of 1.70 GW for steam generation and a concrete storage with a capacity of 40.75 GWh and a maximum heat load of 0.93 GW. Investment costs for the electric boiler and the concrete storage system are 426.14 M€ and 433.49 M€, respectively. Annual energy costs for the optimal electrified system including

thermal energy storage amount to 199.9 M€/y, compared to energy costs of 241.4 M€/y without storage, which corresponds to a saving potential of 17.2%.

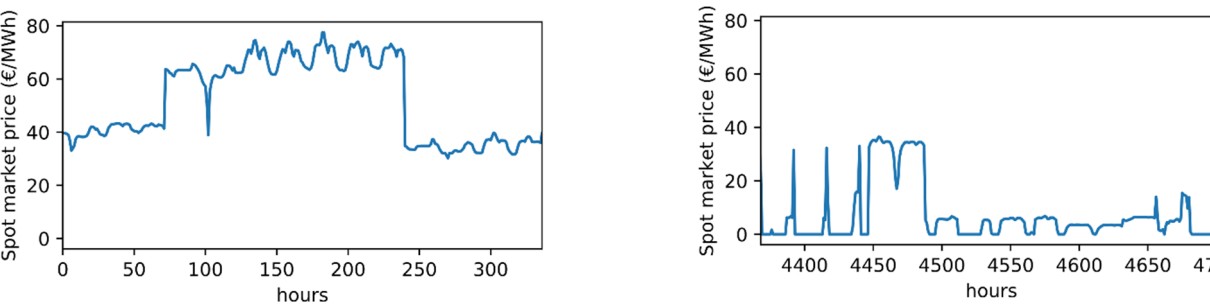

**Figure 8.** Electricity price profiles for representative weeks for dry season (**left**) and wet season (**right**).

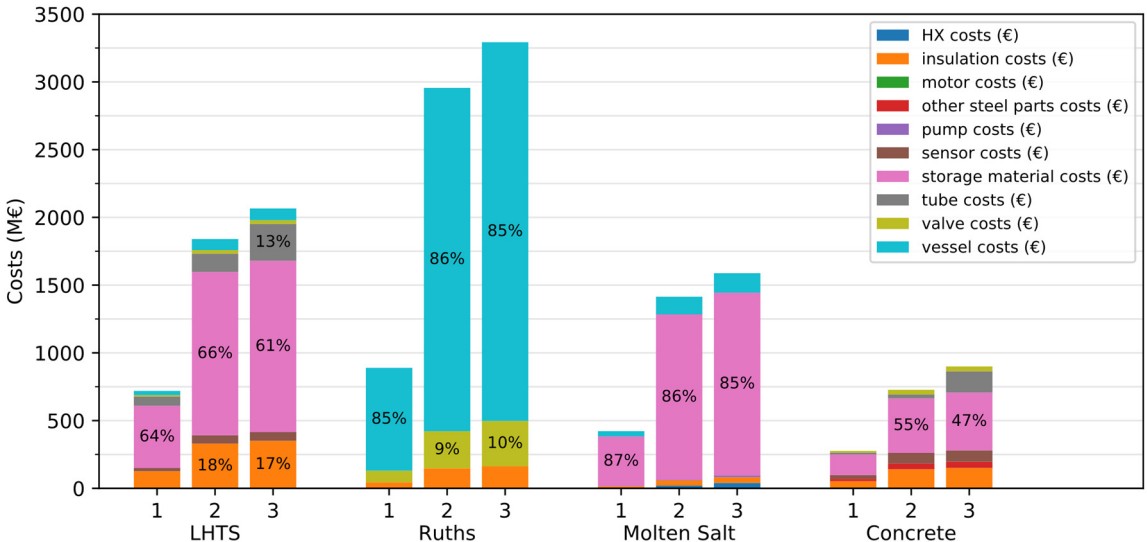

**Figure 9.** Cost structure for all selected TES technologies for Case 1 for three capacity (Cap.)/heat load (HL) scenarios—1: Low Cap./Low HL, 2: High Cap./Low HL, 3: High Cap./High HL.

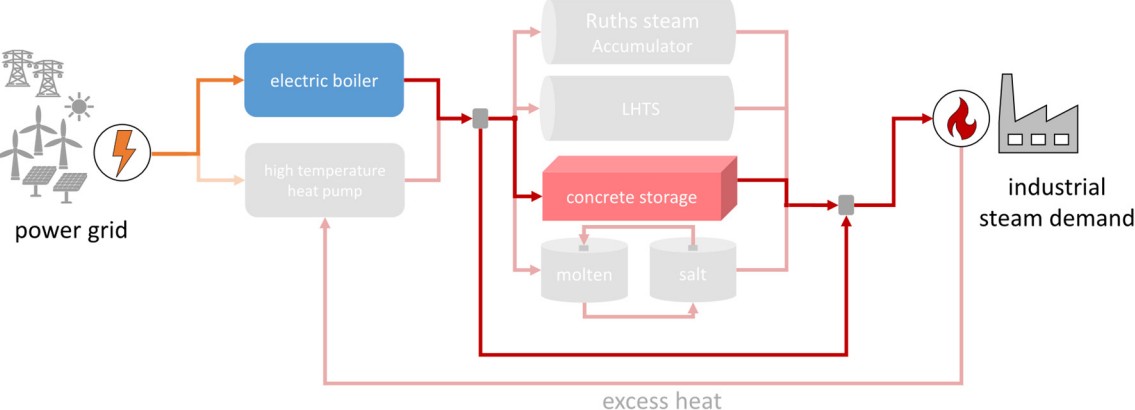

**Figure 10.** Optimal power-to-heat (P2H) system for Case 1.

**Table 4.** Optimal system configuration and resulting costs for Case 1.

| Technology | Max. Heat load (GW) | Capacity (GWh) | Investment Costs (M€) | Energy Costs (M€/y) |
|---|---|---|---|---|
| Electric boiler | 1.70 | - | 426.14 | 199.93 |
| Concrete storage | 0.93 | 40.75 | 433.49 | - |
| Total | - | - | 859.53 | 199.93 |

Base-case (electric boiler only) energy costs: 241.4 M€/y.

Figures 11 and 12 show the boiler heat loads and storage charging (negative values) and discharging rates. As expected, the electric boiler is active in times of relatively low energy prices.

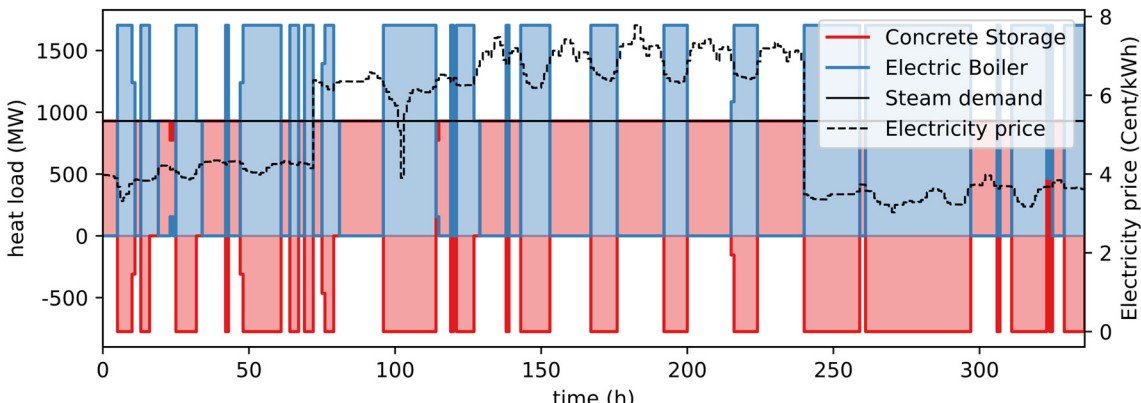

**Figure 11.** Storage and steam generator loads, steam demand and electricity price profiles for Case 1 during dry season.

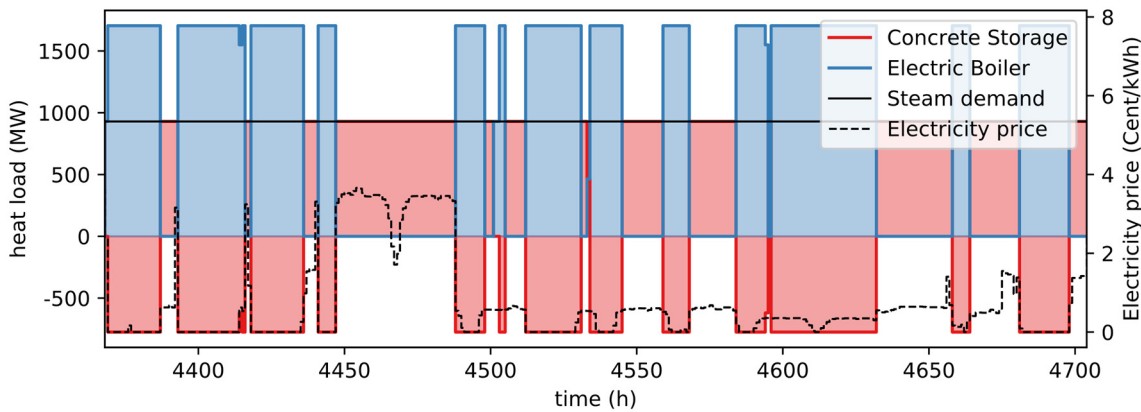

**Figure 12.** Storage and steam generator loads, steam demand and electricity price profiles for Case 1 during wet season.

*5.2. Example Case 2—Medium-Scale Plant with Varying Steam Demand with Low Temperature*

Case 2 represents a central European production facility in the food and beverage sector. The electricity price profile shown in Figure 13 is the real spot market prices from 22 January 2020 for Belgium which, for the sake of simplification, is repeated throughout the entire year. The energy demand in terms of saturated steam shows significant variations throughout the entire period and needs to be supplied at 105 °C. Steam can be produced at temperatures as high as 155 °C which allows for the use of a HTHP. The excess heat factor $f_{surplus}$ is 0.3 and thus 30% of steam supplied to the process can be used by the HTHP as a heat source.

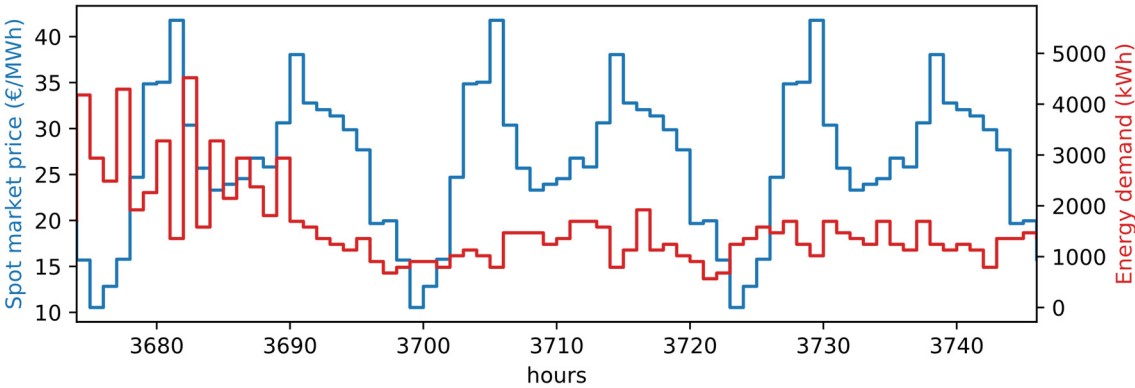

**Figure 13.** Cutout of the electricity price and demand profiles for Case 2.

The storage cost structure for Case 2 presented in Figure 14 is very different compared to Case 1 (Figure 9). LHTS using low-cost high-density polyethylene (HPDE) at a price of 500 €/m³ as a PCM and concrete storage are relatively similar in terms of overall costs. For a combination of high-capacity and low-heat loads (2), storage material costs are the main cost drivers for both LHTS and concrete storage. However, tube costs increase significantly with increased heat load requirements. Costs for Ruths storage are dominated by vessel costs and valve costs, which contribute approximately equally to overall costs. Compared to Case 1, vessel costs are significantly lower due to lower temperature and pressure requirements (Case 2: 155 °C versus Case 1: 300 °C). Molten salt storage is not cost-efficient for Case 2 since costs for storage material are very high. This is due to the salt used, which solidifies at 135 °C and thus only a small temperature range of 20 °C can be used for storage.

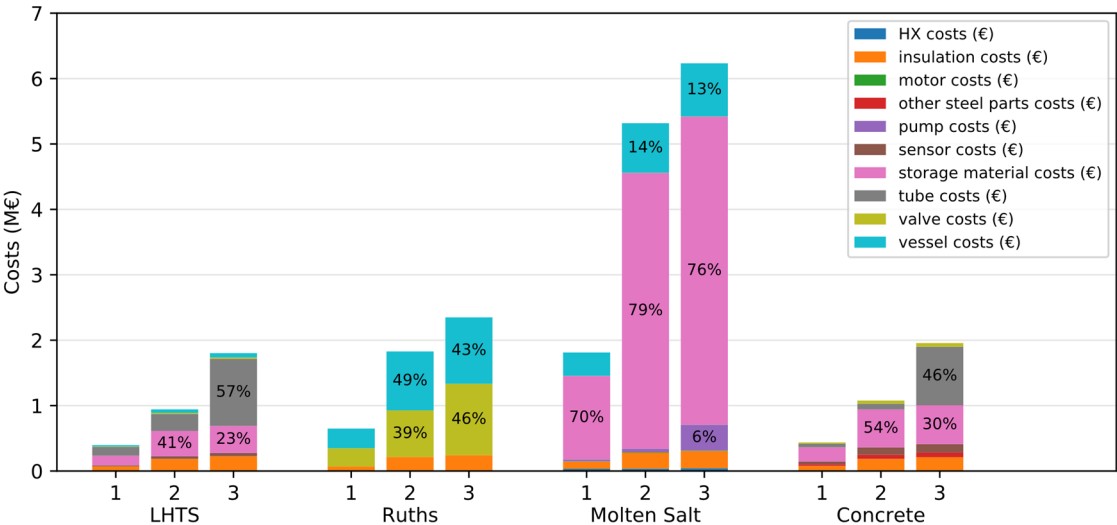

**Figure 14.** Cost structure for all selected TES technologies for Case 2 for three capacity (Cap.)/heat load (HL) scenarios—1: Low Cap./Low HL, 2: High Cap./Low HL, 3: High Cap./High HL.

The optimized system for Case 2, shown in Figure 15 and summarized in Table 5, consists of an electric boiler with a maximum load of 3.8 MW and a high-temperature heat pump with 1.2 MW nominal heat load for steam generation, a concrete storage with a capacity of 1.1 MWh and a maximum heat load of 1.1 MW and an LHTS with a capacity of 13.2 MWh and a maximum heat load of 3.2 MW. Investment costs for the electric boiler and the high-temperature heat pump are 0.95 M€ and 1.22 M€, respectively. Investment costs for the concrete storage are 44.4 k€, and for the LHTS investment costs are 286 k€. Annual energy costs for the optimal electrified system including thermal energy storage amount

to 311 k€/y, compared to energy costs of 476 k€/y without storage. The costs without storage consider steam production using electric boilers. This results in a saving potential of energy costs of 34.7%.

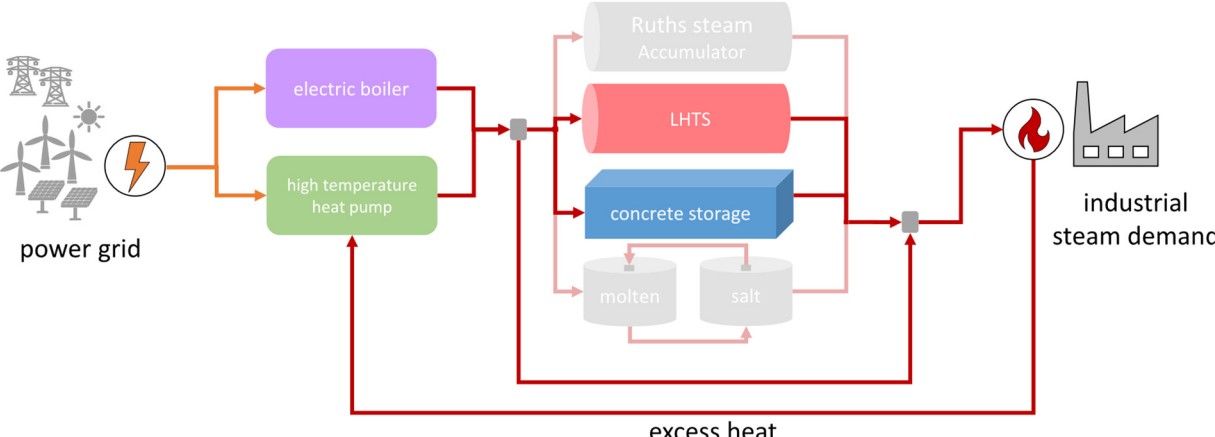

**Figure 15.** Optimal P2H system for Case 2 including electric boilers, high-temperature heat pumps, LHTS and concrete storage.

**Table 5.** Optimal system configuration and resulting costs for Case 2.

| Technology | Max. Heat Load (MW) | Capacity (MWh) | Investment Costs (k€) | Energy Costs (k€/y) |
|---|---|---|---|---|
| Electric boiler | 3.8 | - | 950.0 | 187.8 |
| High-temperature heat pump | 1.2 | - | 1220.0 | 123.0 |
| Latent heat thermal energy storage | 3.2 | 13.2 | 286.0 | - |
| Concrete storage | 1.1 | 1.1 | 44.4 | - |
| Total | - | - | 2505.5 | 310.8 |

Base-case (electric boiler only) energy costs: 476.0 k€/y.

Figure 16 shows a small cutout of the heat load profiles for all components in the P2H system for Case 2. In times of low electricity prices, the electric boiler is used to charge the LHTS, whereas the HTHP is used at more constant heat loads throughout the entire period. The concrete storage seems to be used to reduce peak heat loads of the LHTS.

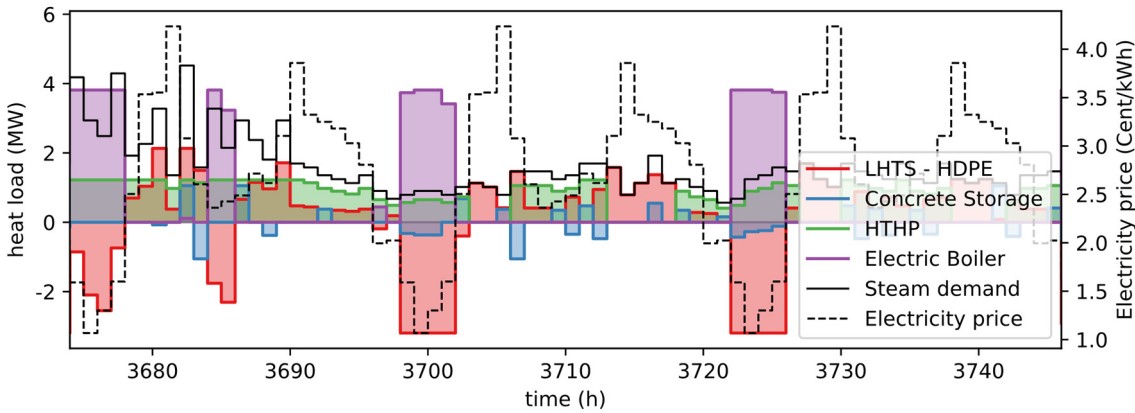

**Figure 16.** Cutout of the thermal storage and steam generator loads, steam demand and electricity price profiles for Case 2.

## 6. Discussion

The proposed optimization approach which consists of the two main modules for cost-function generation and the mathematical programming model allows for detailed cost analysis of the individual TES technologies. At the same time, the approach yields

important decision-support when it comes to selection of cost-efficient TES for The minus sign on the y-axis is related to the font used for plotting in python matplotlib and cannot be easily changed at this point.a specific industrial plant but also to evaluate economic benefits that might emerge from a P2H-system including TES.

The two presented cases and especially the cost structures for the different TES technologies show that case-specific cost estimations with special emphasis on the heat load and temperature requirements is necessary in order to identify the most cost-efficient TES solution. The data presented in Figures 9 and 15 are also shown in Tables 6–8 to facilitate the following discussion.

**Table 6.** Cost structure for Case 1 in percentage of total storage costs (rounded). Dominant cost drivers are printed in bold font. 1: Low Cap./Low HL, 2: High Cap./Low HL, 3: High Cap./High HL.

| | Latent Heat Thermal Energy Storage | | | Ruths Steam Storage | | | Molten Salt Storage | | | Concrete Storage | | |
|---|---|---|---|---|---|---|---|---|---|---|---|---|
| Scenario | 1 | 2 | 3 | 1 | 2 | 3 | 1 | 2 | 3 | 1 | 2 | 3 |
| Heat exchangers (HX) | - | - | - | - | - | - | 1 | 1 | 3 | - | - | - |
| insulation | 18 | 18 | 17 | 5 | 5 | 5 | 3 | 3 | 3 | 19 | 19 | 17 |
| motors | - | - | - | - | - | - | 0 | 0 | 0 | - | - | - |
| other steel parts | - | - | - | - | - | - | - | - | - | 6 | 6 | 5 |
| pumps | - | - | - | - | - | - | 0 | 0 | 1 | - | - | - |
| sensors | 3 | 3 | 3 | - | - | - | 0 | 0 | 0 | 11 | 11 | 9 |
| storage material | **64** | **66** | **61** | - | - | - | **87** | **86** | **85** | **55** | **55** | **47** |
| tubes | 9 | 7 | 13 | - | - | - | - | - | - | 5 | 4 | 17 |
| valves | 1 | 1 | 1 | 10 | 9 | 10 | 0 | 0 | 0 | 5 | 5 | 4 |
| vessels | 4 | 4 | 4 | **85** | **86** | **85** | 9 | 9 | 9 | - | - | - |

**Table 7.** Cost structure for Case 2 in percentage of total storage costs (rounded). Dominant cost drivers are printed in bold font. 1: Low Cap./Low HL, 2: High Cap./Low HL, 3: High Cap./High HL.

| | Latent Heat Thermal Energy Storage | | | Ruths Steam Storage | | | Molten Salt Storage | | | Concrete Storage | | |
|---|---|---|---|---|---|---|---|---|---|---|---|---|
| Scenario | 1 | 2 | 3 | 1 | 2 | 3 | 1 | 2 | 3 | 1 | 2 | 3 |
| Heat exchangers (HX) | - | - | - | - | - | - | 2 | 1 | 1 | - | - | - |
| insulation | 18 | 20 | 13 | 10 | 12 | 10 | 6 | 5 | 4 | 18 | 17 | 11 |
| motors | - | - | - | - | - | - | 0 | 0 | 0 | - | - | - |
| other steel parts | - | - | - | - | - | - | - | - | - | 5 | 6 | 4 |
| pumps | - | - | - | - | - | - | 1 | 1 | 6 | - | - | - |
| sensors | 4 | 4 | 3 | - | - | - | - | - | - | 10 | 10 | 7 |
| storage material | **37** | **41** | 23 | - | - | - | **71** | **79** | **76** | **50** | **54** | 30 |
| tubes | 34 | 27 | **57** | - | - | - | - | - | - | 13 | 8 | **46** |
| valves | 2 | 2 | 1 | 43 | 39 | **46** | - | - | - | 4 | 5 | 3 |
| vessels | 5 | 6 | 4 | **46** | **49** | 43 | 20 | 14 | 13 | - | - | - |

**Table 8.** Average total storage costs in M€ for Case 1 and Case 2. 1: Low Cap./Low HL, 2: High Cap./Low HL, 3: High Cap./High HL.

| | Latent Heat Thermal Energy Storage | | | Ruths Steam Storage | | | Molten Salt Storage | | | Concrete Storage | | |
|---|---|---|---|---|---|---|---|---|---|---|---|---|
| Scenario | 1 | 2 | 3 | 1 | 2 | 3 | 1 | 2 | 3 | 1 | 2 | 3 |
| Case 1 | 719 | 1840 | 2066 | 889 | 2955 | 3293 | 422 | 1414 | 1588 | 276 | 727 | 900 |
| Case 2 | 0.40 | 0.94 | 1.80 | 0.65 | 1.83 | 2.35 | 1.81 | 5.31 | 6.23 | 0.44 | 1.08 | 1.96 |

The available temperature range for storage is especially crucial for the cost-efficient application of Ruths steam storages and LHTS. For Ruths steam storage, vessel costs increase rapidly with higher storage temperatures. This can be observed when comparing the cost structures for Case 1 and Case 2 shown in Tables 6 and 7. For Case 1 with a maximum storage temperature of 300 °C, costs for Ruths storage are dominated by the vessel costs (85–86% of total storage costs) whereas in Case 2 with a maximum storage temperature of 150 °C valves and vessel costs contribute about equally to the total storage costs (Table 7). For LHTS, the

availability of appropriate PCMs with both low costs and high volumetric energy density is a decisive factor regarding cost-effectivity. The volume-specific costs for the selected PCM in Case 2 (LDPE at 500 €/m$^3$) is only half of the costs for Case 1 where a more expensive salt mixture had to be considered (KNO$_3$-NaNO$_3$ at 1000 €/m$^3$).

Case 2 showed that heat load requirements can be a major cost driver for LHTS and concrete storages with approximately doubling costs with doubling maximum heat loads from 0.94 M€ and 1.08 M€ for the High Cap./Low HL scenario to 1.80 M€ and 1.96 M€ for the High Cap./High HL scenario respectively. In Case 2, the relatively low temperature differences for charging and discharging between the HTF and the storage material require large amounts of tubing to establish a sufficient heat transfer. This, in turn, increases the overall volume of the storage and thus increases insulation costs and adds costs for the container structure. In Case 2, for both LHTS and concrete storage tubing makes up for 57% and 46% of the overall storage costs in the High Cap./High HL scenario (Table 7) compared to Case 1 where in the High Cap./High HL scenario tubing costs make up for only 13.0% in the case of LHTS and 11.6% for the concrete storage (Table 6).

Some cost drivers considered within the cost functions showed only minor impact to the overall storage costs; e.g. heat exchanger (HX) costs that were considered for molten salt storage made up for a maximum of only 3%. The motors considered for pumping of liquid salt showed even less impact with less than 0.5% of total storage costs. The molten salt costs were in both cases dominated by the storage material costs.

In the proposed approaches for cost-function generation, some aspects that might have a significant effect on costs, were not fully considered. Economy of scales was only considered for steel plates but was not applied for storage material costs. For large-scale applications such as Case 1, this effect might change the cost structure of the individual storage, as well as the choice of cheapest storage technology. In Case 1, the storage material was responsible for 50–85% of the total TES costs for molten salt, LHTS and concrete TES. This aspect, however, can be included and does not change the effectiveness of the proposed optimization approach.

Controllability of storage heat loads, which is another important aspect, was not considered in detail, but instead perfect control over charging and discharging heat loads was assumed. For a more detailed analysis, transient storage simulations will be necessary to fully evaluate, whether the individual storage technology can fulfil all process requirements.

One major limitation of the proposed approach is that heat loads considered for LHTS and concrete are average values obtained from simulation of a full charging cycle. Heat load restrictions depending on the state of charge cannot be considered as this would yield a non-linear storage model which would be very difficult to solve. The presented approach underestimates initial maximum heat loads of LHTS and concrete storage and overestimates obtainable heat loads at higher (charging) or lower (discharging) levels of SOC.

There are also other minor shortcomings in the present model that could be addressed in future work:

- A constant heat transfer coefficient was assumed for LHTS and concrete storages;
- Preheating of makeup water and condensate was not considered;
- Heat losses are neglected;
- PCM selection for LHTS is not automated (manual selection of appropriate PCM);
- Automated sensitivity analysis (sensitivity regarding storage costs);
- Economy of scale is not considered for storage materials.

## 7. Conclusions

Application of high-temperature TES in steam production is expected to become increasingly relevant to enable decarbonization of the process industry with increased share of fluctuating renewable energy sources in the grid. The present paper demonstrates that heat load-specific costs must not be neglected when it comes to cost-optimal storage selection for high temperature applications such as industrial steam supply, since heat load requirements usually have a significant impact on heat transfer areas. This is especially true

in the case of indirect thermal energy storage by means of an intermediate storage medium. The derived storage cost functions are not only capacity but also heat load-dependent which is crucial for industrial applications. Moreover, cost optimal storage integration for industrial storage applications at higher temperatures (>100 °C) has only been addressed by a few authors. The proposed optimization model can easily be extended for other steam generation units and storages since its formulation is general. The characteristics of the different storage technologies are considered by means of parameter values. Using a linear approximation of storage costs with respect to storage capacity and heat load, the solution for the optimization problem can be obtained within seconds or minutes considering electricity and demand profiles for one year, which yields a very promising basis for a potential decision support tool.

**Author Contributions:** Conceptualization, A.B., G.D.-S., H.K. and A.S.; methodology, A.B.; software, A.B., H.K., M.S. and A.S.; validation, A.B., H.K. and A.S.; formal analysis, H.K. and A.S.; investigation, A.B., H.K. and A.S.; resources, A.B., G.D.-S., H.K., M.S. and A.S.; data curation, A.B., H.K. and A.S.; writing—original draft preparation, A.B.; writing—review and editing, H.K. and A.S.; visualization, A.B.; supervision, H.K. and G.DS; project administration, H.K. and A.B.; funding acquisition, H.K. and G.DS.. All authors have read and agreed to the published version of the manuscript.

**Funding:** The research leading to this publication has been funded by HighEFF—Centre for an Energy Efficient and Competitive Industry for the Future, an 8-year Research Centre under the FME-scheme (Centre for Environment-friendly Energy Research, 257632). The authors gratefully acknowledge the financial support from the Research Council of Norway and user partners of HighEFF.

**Institutional Review Board Statement:** Not applicable.

**Informed Consent Statement:** Not applicable.

**Data Availability Statement:** Restrictions apply to the availability of some of the cost information used for this study. Cost data was taken from the DACE price booklet and are available at dacepricebooklet.com/.

**Acknowledgments:** The authors acknowledge the user partners of HighEFF for contributing with cost information and relevant cases for the study.

**Conflicts of Interest:** The authors declare no conflict of interest.

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
