# Peer review of "Optimal Selection of Thermal Energy Storage Technology for Fossil-Free Steam Production in the Processing Industry"

_applsci, doi:10.3390/app11031063_

Round 1
Reviewer 1 Report
Overall, Figures are good and help to understand better subject and results, figures 10 and 15 masters the concept
However, the following corrections are required:
- Line 53: Explain shortly what are spot market and reserve markt
- Line 59: Why did you choose for "particle swarm optimization (PSO)"? Explain the methodology in short terms.
- Line 91: Do you mean with specific storage capacity costs? Does it include the material type?
- Lines 92 and 108: Improve the English, not only - but also is not correctly implemented which makes difficult the understanding of concept.
- Line 131: What do you mean with "low energy density"? Availability or requirements?
- Line 150: requires a reference for the claimed linear function
- Line 162: What do you mean with "demand and excess heat only occur simultaneously"? It does not make sense to be correct always.
- Line 206: How are these costs considered in annual base? Explain more about fa
- Line 226: Format and language is not correct: Table 1Fehler! Ungültiger Eigenverweis auf Textmarke..
(This is not acceptable)
- Line 227: In table 1, specify what are empty places
- Line 245: What is the mentioned data base? Experimental or estimated values? cost database for cylindrical pressure
- Line 252: Why did you fixed the maximum flow rate within the inlet and outlet of the vessel is set to 20 and 25 m/s? Introduce the reference as well
- Line 295: How much is Δ???? considered? And the reason is required
- Line 344: what is the basis of estimate? any reference?
- Line 369: The tank thickness of 10 mm cannot be always acceptable. You should consider standers and possible changes at thickness
- Page 17: Put the explained data in a table and go through analyzing and reasons in the text.
- Line 523: much more is not correct! You can use even more...
- At the end, reviewer suggest some more cases for the lower and higher heat loads and reporting of the results in a table format which makes the comparison easier.
Author Response
We would like to thank the reviewer for his/her thorough review and appreciate his/her constructive and positive feedback. In the following you will find our comments. Our responses are right below the reviewer's comment in italic font.
-------------
Overall, Figures are good and help to understand better subject and results, figures 10 and 15 masters the concept
However, the following corrections are required:
- Line 53: Explain shortly what are spot market and reserve market
Old: “The objective in this case was to optimize energy planning and trading within distributed energy systems, also targeting spot market and reserve market participation.”
New: “The objective in this case was to optimize energy planning and trading within distributed energy systems, also targeting short term trades at the spot market and participation at the reserve market providing balancing power.”
- Line 59: Why did you choose for "particle swarm optimization (PSO)"? Explain the methodology in short terms.
It was not us that used PSO, this was a part of the literature survey.
We hope this slight change in grammar makes it clearer now.
Old: “Capacity models were also used for the optimization of a tri-generation system including TES using particle swarm optimization (PSO) [5], within a simple storage model for optimization of a poly-generation district energy system [6], and for optimization including a simple ice storage with loss free heat transfer [7].”
New: “Capacity models have also been used for the optimization of a tri-generation system including TES using particle swarm optimization (PSO) [5], within a simple storage model for optimization of a poly-generation district energy system [6], and for optimization including a simple ice storage with loss free heat transfer [7].”
- Line 91: Do you mean with specific storage capacity costs? Does it include the material type?
The capacity specific storage costs are the total storage costs per unit of energy content (e.g. €/kWh). It includes all the costs of the storage.
Old: “For example, for Ruths steam storages, the applicable temperature range and especially the maximum allowable storage temperature and pressure both influence the specific storage capacity in terms of energy content, but also the capacity specific storage costs.”
New: “For example, for Ruths steam storages, the applicable temperature range and especially the maximum allowable storage temperature and pressure both influence the volume- and mass-specific storage capacity in terms of energy content, but also the capacity specific storage costs. The capacity specific storage costs are the total storage costs per unit of energy content (e.g. €/kWh).”
- Lines 92 and 108: Improve the English, not only - but also is not correctly implemented which makes difficult the understanding of concept.
Old: “Higher storage pressures result in thicker pressure vessels to contain increased internal pressures, but also steel strength decreases with increased pressures and temperatures.”
Old: “This includes the optimal storage capacity and the required heat loads but also optimal storage operation.”
New: “Higher storage pressures not only result in thicker pressure vessels to contain increased internal pressures, but also reduced steel strength due to increased pressures and temperatures.”
New: “This not only includes the optimal storage capacity and the required heat loads but also optimal storage operation.”
- Line 131: What do you mean with "low energy density"? Availability or requirements?
Old:” … but the technology is limited by the low energy density”
New:” … but the technology is limited by its relatively low energy density compared to e.g. LHTS.”
- Line 150: requires a reference for the claimed linear function
Old:” The investment costs for electric boilers C_invest^B are a linear function of the maximum heat load Q ̇^(B,max) with the cost coefficients c_0^B and c_1^B.”
New:” For simplicity, the investment costs for electric boilers C_invest^B were considered to be a linear function of the maximum heat load Q ̇^(B,max) with the cost coefficients c_0^B and c_1^B.”
- Line 162: What do you mean with "demand and excess heat only occur simultaneously"? It does not make sense to be correct always.
The idea is that the excess heat stems from the plants internal processes requiring heat, and thus the availability of excess heat is limited to periods when there is a heat demand. For simplicity, thermal inertia was neglected.
Old:” It is assumed that only a fraction of the process’ heat demand is available as excess heat and that demand and excess heat only occur simultaneously.”
New:” It is assumed that only a fraction of the process’ heat demand is available as excess heat, and that there is excess heat available only when there is a heat demand. For simplicity, thermal inertia was neglected. However, the model can easily be modified if actual excess heat profiles are available.”
- Line 206: How are these costs considered in annual base? Explain more about fa
Old:” To consider energy and investment costs on the same basis, the annualization factor f_a is used and corresponds in this case to the equipment’s life expectancy.”
New:” To consider energy and investment costs on the same basis, the annualization factor f_a is used to calculate annuities for the investments. In this case f_a corresponds to the reciprocal of the equipment’s life expectancy.”
- Line 226: Format and language is not correct: Table 1Fehler! Ungültiger Eigenverweis auf Textmarke..
(This is not acceptable)
The flawed reference was corrected.
- Line 227: In table 1, specify what are empty places
Empty spaces show that the components were not considered for the respective storage technology
The table caption was changed:
Old:” Components and key variables considered with respect to selected TES technologies”
New:” Components and key variables that impact the respective component costs for the selected TES technologies”
- Line 245: What is the mentioned data base? Experimental or estimated values? cost database for cylindrical pressure
The data base is the DACE price booklet (ref. [18])
The reference was added in the text.
- Line 252: Why did you fixed the maximum flow rate within the inlet and outlet of the vessel is set to 20 and 25 m/s? Introduce the reference as well
We introduced a reference to a guide published by SPIRAX SARCO that states that the maximum flowrate for saturated steam should not exceed 25 m/s.
We added the following sentence:
“This is slightly lower than the limits of 25 m/s for saturated steam (outlet) and 40-60 m/s for dry steam (inlet) as suggested in literature [22]“
- Line 295: How much is Δ???? considered? And the reason is required
We added the following sentence:
“…, which was set to 0.8 in this work. This factor reduces the theoretically available temperature range to a more realistic range where reasonable driving temperature differences are ensured.”
- Line 344: what is the basis of estimate? any reference?
We based these estimates on previously conducted projects, where sensors were actually purchased. The text was changed accordingly.
- Line 369: The tank thickness of 10 mm cannot be always acceptable. You should consider standers and possible changes at thickness
Old:” The tank thickness was set to a constant value of 10 mm.”
New:” Since the storage tanks are under atmospheric pressure, the tank thickness was set to a constant value of 10 mm, even though in certain cases thicker walls might be necessary.”
- Page 17: Put the explained data in a table and go through analyzing and reasons in the text.
As suggested, we put the resulting optimal system configurations into tables.
- Line 523: much more is not correct! You can use even more...
Statement was changed to “…very difficult to solve.”
- At the end, reviewer suggest some more cases for the lower and higher heat loads and reporting of the results in a table format which makes the comparison easier.
We feel that showing these two examples highlights the main issue we wanted to address. That is the temperature and heat load dependency of the storage costs and that using fixed capacity specific storage costs is not sufficient when it comes to cost-based optimization.
However, we welcome the suggestion of putting the results for the optimal system configurations into tables.

Reviewer 2 Report
A way to produce steam at high efficiency, high quality and at high Carnot factor is given by the use of Solid Oxide Fuel Cell.
In this regards it is suggested to include in the Introduction papers treating SOFC technology. Here some suggestions:
- https://doi.org/10.18280/ijht.34S217
- https://doi.org/10.1016/j.enconman.2020.112664
The reviewer suggests the authors to include some numerical values to compare the technical solutions in the discussion paragraph.
Author Response
We would like to thank the reviewer for his/her review and appreciate his/her constructive and positive feedback. In the following you will find our comments. Our responses are right below each reviewer’s comment and in italic font.
The changes in the manuscript are included in review mode.
-----------
Comments and Suggestions for Authors
A way to produce steam at high efficiency, high quality and at high Carnot factor is given by the use of Solid Oxide Fuel Cell.
In this regards it is suggested to include in the Introduction papers treating SOFC technology. Here some suggestions:
- https://doi.org/10.18280/ijht.34S217
- https://doi.org/10.1016/j.enconman.2020.112664
We agree that SOFC can be a promising solution for CHP applications and steam production. The potential use of hydrogen as a future energy carrier could be interesting with regards to electrification of industrial energy supply (power 2 hydrogen 2 power&heat). However, we think that this route is out of scope for our publication since we focus on thermal energy storage and steam production with renewable electricity. The actual steam production is however only considered in a very simplified way (basic models for boilers and high temperature heat pumps).
The reviewer suggests the authors to include some numerical values to compare the technical solutions in the discussion paragraph.
Some numerical values were added:
“In Case 2, for both LHTS and concrete storage tubing makes up for about half of the overall storage costs in the High Cap. / High HL-case compared to Case 1 where in the High Cap. / High HL-case tubing costs make up for only 13 % in the case of LHTS and 11.6% for the concrete storage.”
“In Case 1, the storage material was responsible for 50-85 % of the total TES costs for molten salt, LHTS and concrete TES.”

Reviewer 3 Report
This article dealing with thermal energy storage is interesting. It is well written and could be published in this journal if the authors consider the following comments considered minor:1. complete the article with some passive energy storage systems used in building such as the Trombe Wall
2. make a general conclusion of the article
Author Response
We would like to thank the reviewer for his/her review and appreciate his/her constructive and positive feedback. In the following you will find our comments. Our responses are right below each reviewer’s comment and in italic font.
The changes in the manuscript are included in review mode.
----------
Comments and Suggestions for Authors
This article dealing with thermal energy storage is interesting. It is well written and could be published in this journal if the authors consider the following comments considered minor:
1. complete the article with some passive energy storage systems used in building such as the Trombe Wall
Thank you for your suggestion. We do think that energy storages used in building applications are not as relevant when it comes to stream production for industrial applications, since temperature levels are much lower for heating purposes and thermal management. This is why we put our focus on storage technologies with storage temperatures > 100°C.
- make a general conclusion of the article
This paragraph was added to conclude the article:
“The present paper demonstrates that heat load specific costs must not be neglected when it comes to cost-optimal storage selection for high temperature applications such as industrial steam supply, since heat load requirements usually have a significant impact on heat transfer areas. This is especially true in case of indirect thermal energy storage by means of an intermediate storage medium. The derived storage cost functions are not only capacity but also heat load dependent which is crucial for industrial applications. Moreover, cost-optimal storage integration for industrial storage applications at higher temperatures (>100°C) has only been addressed by few authors in general.
The proposed optimization model can easily be extended for other steam generation units and storages since its formulation is general. The characteristics of the different storage technologies are considered by means of parameter values. Using a linear approximation of storage costs with respect to storage capacity and heat load, the solution for the optimization problem can be obtained within seconds or minutes considering electricity and demand profiles for one year, which yields a very promising basis for a potential decision support tool.”

Round 2
Reviewer 1 Report
Attached please find the comments. Red and yellow highlighted sentences show the required changes/improvements
